# Antennal Transcriptome Analysis and Identification of Olfactory Genes in *Glenea cantor* Fabricius (Cerambycidae: Lamiinae)

**DOI:** 10.3390/insects13060553

**Published:** 2022-06-17

**Authors:** Guanxin Wu, Ranran Su, Huili Ouyang, Xialin Zheng, Wen Lu, Xiaoyun Wang

**Affiliations:** Guangxi Key Laboratory of Agric-Environment and Agric-Products Safety, College of Agriculture, Guangxi University, Nanning 530004, China; wugx0623@163.com (G.W.); srr18838933489@126.com (R.S.); oyhl050023@163.com (H.O.); zheng-xia-lin@163.com (X.Z.); luwenlwen@163.com (W.L.)

**Keywords:** *Glenea cantor*, antennal transcriptome, olfactory genes, phylogenetic tree, expression patterns

## Abstract

**Simple Summary:**

In this study, we conducted antennal transcriptome analysis in *Glenea cantor* (Cerambycidae: Lamiinae) and identified 76 olfactory-related genes, including 29 odorant binding proteins (OBPs), 14 chemosensory proteins (CSPs), 13 odorant receptors (ORs), 18 ionotropic receptors (IRs) and 2 sensory neuron membrane proteins (SNMPs). We also verified the reliability of transcriptome differential genes by qRT-PCR, which indicated the reliability of the transcriptome. Based on the relative expression of 30 d adults, GcanOBP22 and GcanOBP25 were highly expressed not only in the antennae, but also in the wings and legs. In addition, GcanCSP4 was the highest expression on the female antennae at 12 d. These findings laid the foundation for further research on the mechanism of *G. cantor* olfactory mechanism at the molecular level.

**Abstract:**

*Glenea cantor* Fabricius (Cerambycidae: Lamiinae) is a pest that devastates urban landscapes and causes ecological loss in southern China and Southeast Asian countries where its main host kapok trees are planted. The olfactory system plays a vital role in mating, foraging, and spawning in *G. cantor* as an ideal target for pest control. However, the olfactory mechanism of *G. cantor* is poorly understood at the molecular level. In this study, we first established the antennal transcriptome of *G. cantor* and identified 76 olfactory-related genes, including 29 odorant binding proteins (OBPs), 14 chemosensory proteins (CSPs), 13 odorant receptors (ORs), 18 ionotropic receptors (IRs) and 2 sensory neuron membrane proteins (SNMPs). Furthermore, the phylogenetic trees of olfactory genes were constructed to study the homology with other species of insects. We also verified the reliability of transcriptome differential genes by qRT-PCR, which indicated the reliability of the transcriptome. Based on the relative expression of 30 d adults, GcanOBP22 and GcanOBP25 were highly expressed not only in the antennae, but also in the wings and legs. In addition, GcanCSP4 was the highest expression on the female antennae at 12 d. These findings laid the foundation for further research on the mechanism of *G. cantor* olfactory mechanism at the molecular level.

## 1. Introduction

The olfactory system of insects is important to the completion of host positioning, feeding, mating, oviposition, and other behaviors [1,2]. Insects can perceive the information of odor molecules in the external environment and respond accordingly [3], and primarily rely on the antennae, a primary chemical sensory organ with various sensilla [4]. In general, the olfactory system mainly involves several gene families: odorant binding proteins (OBPs), chemosensory proteins (CSPs), odorant receptors (ORs), ionotropic receptors (IRs) and sensory neuron membrane proteins (SNMPs), which mainly locate on the sensilla to process various odorants or pheromones and elicit related behaviors [5,6].

Among these gene families, OBPs are a class of small soluble proteins widely present in the lymph fluid of the antennal sensilla, which can bind and transfer such molecules to the receptors [7], and participate in host location and pheromone perception [8,9]. The combination of OBPs and volatile molecules is the foremost step for insects to recognize odorous pheromones [10,11]. CSPs consist of a class of small conserved binding proteins, whose functions are similar to that of OBPs, but they bind more odorants than OBPs [12]. In addition, CSPs have broader functions, such as binding nutrients and hormones [13].

ORs are a class of receptor proteins, which have a seven-transmembrane domain and mainly recognize odor molecules, including general odors and sex pheromones from the natural environment [14,15]. In addition, ORs are dimerized with OR co-receptor (ORco) during the formation of odorant-gated ion channels, which can help the positioning accuracy of typical ORs, thereby increasing the sensitivity to odor molecules [16]. Pheromone receptors (PRs) are also a subfamily of ORs that specializes in detecting pheromones and analogs [17]. IRs are another kind of chemical receptor protein, and they are related to ionotropic glutamate receptors (iGluR), which play a role in odor-induced activation responses [18]. SNMPs are related to scavenger proteins of the CD36 protein family [19]. In general, SNMPs can be divided into two types: SNMP1 plays a role in the response to pheromone, which is located on neuronal membranes [20]; SNMP2 is primarily expressed in the support cells or pheromone-sensitive sensilla neurons, but their potential function remains unknown [21].

*Glenea cantor* Fabricius (Cerambycidae: Lamiinae) (Appendix A) is an important pest that is destructive to kapok *Bombax ceiba* Linnacus, which is widely distributed in China and Southeast Asian countries [22,23]. *G. cantor* is oligophagous and consumes almost exclusively kapok branches [24]. In severe cases, it usually leads to the continuous death of kapok trees, which seriously affects urban greening and landscaping. In Nanning, China, there are four generations of *G. cantor* a year, 70 days per generation [25,26]. At present, considerable research about *G. cantor* focuses on its biological characteristics [26,27,28], behavioristics [22,23], prevention and control [24], artificial breeding method [22,29], mitochondrial genome [30] and reference genes [31]. As for the olfactory system, Dong et al. observed the ultrastructure of adult *G. cantor* antennae, which was the only published report [28]. We have also discovered several volatiles from kapok, which play a role in host orientation in *G. cantor* (unpublished data). There is a phenomenon of overlapping generations of *G. cantor*. In addition, the kapok tree is tall, and it is difficult to obtain the desired effect by chemical control. At present, the control methods are mainly to strengthen the management of trees, and capture manually and trap adults [24]. Thus, understanding of the olfactory system of *G. cantor* remains inadequate because of a lack of basic olfactory genetic background.

Here, we constructed the antennal transcriptome database of *G. cantor* and identified olfactory gene clusters by transcriptome analysis. In addition, we analyzed the phylogenetic relationship between these genes and orthologs from other species, and verified the reliability of the transcriptome and expression patterns. These results should lay the foundation for an in-depth understanding of the olfactory mechanism of *G. cantor* and provide a reference for further research on the prevention and treatment of longhorn beetles through olfactory mechanisms.

## 2. Materials and Methods

### 2.1. Insects and Sample Collection

The adult *G. cantor* used in this study were obtained by long-term laboratory rearing at Guangxi University, China (25 ± 1 °C, 75 ± 5% humidity, 14 L: 10 D) [28]. Adults were fed on healthy kapok branches. The antennae of healthy male and female adults were selected and divided into groups (a group of five adults). We cut out the adult antennae of 1 d, 5 d, 12 d, 20 d, and 30 d as age expression. Tissue expression was obtained from 30 d adults and divided into the antennae, head (without antennae), thorax, abdomen, legs, and wings. Three biological replicates were performed per group. Samples were excised on ice and immediately frozen in liquid nitrogen, and stored at −80 °C until RNA isolation.

### 2.2. RNA Sequencing

Total RNA of mixed age from respectively 10 female antennae and 10 male antennae per replicate (three biological replicates) was extracted following the manufacturer’s protocol by using the Trizol reagent kit (Invitrogen, Carlsbad, CA, USA). Eukaryotic messenger RNA (mRNA) was enriched from the total RNA by Oligo (dT) beads. Then, cDNA fragments were purified using the QiaQuick PCR extraction kit (Qiagen, Venlo, The Netherlands), ends repaired, A bases added, and ligated to Illumina sequencing adapters. The ligation products were screened by agarose gel electrophoresis and amplified by PCR and RNA sequencing was performed using novaseq 6000 (Illumina) by Gene Denovo Biotechnology Co. (Guangzhou, China).

### 2.3. Transcriptome Assembly, Annotation of Unigenes, and DGE Analysis

Reads obtained from the sequencing machines included raw reads containing adapters or low-quality bases which would affect the following assembly and analysis. Thus, to obtain high-quality clean reads, reads were further filtered by fastp (version 0.18.0). The parameters were as follows: (1) removing reads containing adapters; (2) removing reads containing more than 10% of unknown nucleotides (N); (3) removing low quality reads containing more than 50% of low quality (Q-value ≤ 20) bases.

Transcriptome de novo assembly was performed using the Trinity method [32]. Unigenes were annotated by using the BLASTx project (http://www.ncbi.nlm.nih.gov/BLAST/, accessed on 11 October 2021) with an E-value threshold of 1 × 10^5^ to the NCBI non-redundant protein (Nr) database (http://www.ncbi.nlm.nih.gov, accessed on 11 October 2021), Swiss-Prot protein database (http://www.expasy.ch/spro-t, 11 October 2021), Kyoto Encyclopedia of Genes and Genomes (KEGG) database (http://www.genome.jp/kegg, accessed on 11 October 2021), and COG/KOG database (http://www.ncbi.nlm.nih.gov/COG, accessed on 11 October 2021). Then protein functional annotations were obtained on the basis of the best alignment results. The different gene expressions (DGEs) were calculated and normalized to RPKM (Reads Per kb per Million reads) and the relative expressions of differential expressed genes were viewed by volcano plot.

### 2.4. Identification of Olfactory Genes

The tBLASTn algorithm was used to identify cDNA sequences encoding candidate OBP, CSP, OR, IR, and SNMP genes of *G. cantor* from the obtained transcriptome database. Geneious Prime (2022.1 for windows, Auckland, New Zealand) was also used to fill possible omissions by local BLAST. Reference sequences from other coleoptera were downloaded from NCBI, including *Agrilus mali*, *Anomala corpulenta*, *Anoplophora glabripennis*, *Apriona germari*, *Batocera horsfieldi*, *Colaphellus bowringi*, *Dendroctonus adjunctus*, *Dendroctonus armandi*, *Dendroctonus ponderosae*, *Eucryptorrhynchus scrobiculatus*, *Holotrichia parallela*, *Hycleus cichorii*, *Ips typographus*, *Leptinotarsa decemlineata*, *Megacyllene caryae*, *Monochamus alternatus*, *Ostrinia nubilalis*, *Phyllotreta striolata*, *Propsilocerus akamusi*, *Pyrrhalta aenescens*, *Pyrrhalta maculicollis*, *Rhynchophorus ferrugineus*, *Rhyzopertha dominica*, *Tenebrio molitor*, *Tribolium castaneum* and *Xylotrechus quadripes*.

### 2.5. Sequence and Phylogenetic Analysis

Signal peptides were calculated by signalP 4.1 (http://www.cbs.dtu.dk/services/SignalP/, accessed on 15 October 2021). TransMembrane prediction was performed by using the Hidden Markov Model 2.0 (http://www.cbs.dtu.dk/services/TMHMM, accessed on 15 October 2021). Amino acid sequence alignment was performed by using the ClustalW method in MEGA software (version 7.0, Auckland, New Zealand), and phylogenetic trees were built by using an online tool (IQ-TREE; http://iqtree.cibiv.univie.ac.at/, accessed on 18 October 2021). Moreover, all trees were drawn by using FigTree v1.4.3 (http://tree.bio.ed.ac.uk/software/figtree/, accessed on 18 October 2021) and Adobe Illustrator CC 2018 (Adobe, CA, USA).

### 2.6. Quantitative Real-Time PCR (qRT-PCR)

We randomly selected eight unigenes related to olfactory functions (OBP22, OBP25, CSP4, IR6, IR8-10, IR18) to validate the accuracy of the transcriptome results by qRT-PCR. Primers (Appendix A) were designed by NCBI (https://www.ncbi.nlm.nih.gov/tools/primer-blast/, accessed on 25 October 2021), and sequenced after PCR. To validate the transcriptome, we performed qRT-PCR experiments with the same sequenced RNA samples. Input RNA was used as a template for cDNA synthesis, which was synthesized following the instructions of the PrimeScript RT Reagent Kit with gDNA Eraser (Takara, Dalian, China). qRT-PCR was performed on a LightCycler (F. Hoffmann-La Roche Ltd., Basel, Switzerland) using TB Green Premix Ex Taq™ (Takara, Dalian, China). qRT-PCR conditions were set as follows: 40 cycles of 95 °C for 5 s and 60 °C for 34 s.

Age and tissue expressions of OBP22, OBP25, and CSP4 were further conducted. The RNAiso Plus reagent (Takara, Dalian, China) was used to obtain total RNA from different ages and tissues including respectively 10 antennae, 5 heads (without antennae), 5 thoraxes, 5 abdomens, legs of 5 adults, and wings of 5 adults per replicate (three biological replicates), which were isolated from 30-day-old healthy female and male adults and respectively 10 antennae per replicate (three biological replicates) from 1 d, 5 d, 12 d, 20 d and 30 d of female and male. Whether RNA from other tissues is also extracted in the same way as the antennae and the experimental procedure was the same as above. In this study, *RPL32* and *EF1A1* were chosen as the reference genes in qRT-PCR assays to standardize the tissue’s expression levels of each gene, while *RPL36* and *EF1A1* were chosen as the reference genes to age expression levels [33]. In addition, the calibration curve was constructed by gradient diluting each cDNA template and R^2^ value > 0.990 (Appendix A). The slope of the standard curve was used to calculate the amplification efficiency of each primer pair, and the efficiencies of them were between 90% and 105% (Appendix A). Meanwhile, the melting curve analysis displayed a single sharp peak from 83 to 88 °C (Appendix A). The relative expression level was calculated by using the 2^−ΔΔCt^ method with three biological replicates and three technical replicates [33]. Image deconvolution and quality value calculations were performed using GraphPad Prism 8 (Inc., La Jolla, CA, USA) and Adobe Photoshop CC 2018 (Adobe, CA, USA).

## 3. Results

### 3.1. Overview of G. cantor Antennal Transcriptome

The female *G. cantor* antennal transcriptome yielded 40,989,086 raw reads and 38,130,472 clean reads, of which Q20 accounted for 99.78%. In addition, the antennal transcriptome of male *G. cantor* yielded 42,482,724 raw reads and 42,395,862 clean reads, of which Q20 accounted for 99.80%. The antenna-specific assembly produced 24,462 unigenes, with GC value of 44.18%. The number of N50 was 8089 from 116 bp to 12,058 bp (Table 1).

Gene Ontology (GO) annotation analysis showed that 289,811 unigenes could be annotated into three functional categories, namely biological process, cellular component, and molecular function (Figure 1). Moreover, KEGG pathway annotation analysis showed that 28,541 genes could be annotated into six KEGG categories (Appendix A), namely metabolism (6744 unigenes; 23.70% of the unigenes annotated to the KEGG database), genetic information processing (2121; 7.43%), environmental information processing (2937; 10.29%), cellular processes (3450; 12.09%), organismal systems (6270; 21.97%) and human diseases (7019; 24.59%). Among the 46 subcategories, the global and overview maps were the subcategory annotating the largest number of genes, followed by signal transduction and infectious. COG/KOG analysis showed that a total of 24,448 unigene sequences were classified into 25 COG functional categories. The largest category was the general function prediction only group (4015; 16.42%), followed by the signal transduction mechanisms group (3863; 15.80%), posttranslational modification, protein turnover, chaperones group (1913; 7.82%), as well as the cytoskeleton group (1465; 6.00%) (Appendix A). A total of 488 differentially expressed genes were identified in the antennal transcriptome comparing sexes, of which 268 unigenes were upregulated and 220 unigenes were downregulated (Appendix A). By comparing the FPKM values, it was found that there were 10 olfactory genes which were sex-biased from the transcriptome. Among them, OBP22, OBP25, CSP4, and CSP10 were female-biased, IR6, IR8, IR9, IR10, IR15, and IR18 were male-biased.

### 3.2. Identification of Candidate OBPs

A total of 29 OBPs were identified in the *G. cantor* antennal transcriptome. The results showed that all GcanOBPs had complete open reading frames (ORFs) encoded from 62 to 247 amino acid residues and revealed that four GcanOBPs (GcanOBP8, 17, 20, and 29) contained six conserved cysteine residues, which were typical structures of insect OBPs. In addition, 17 GcanOBPs (GcanOBP1-2, 4, 6, 10–12, 15–16, 18, 21–26, and 28) were characterized by minus-C and GcanOBP3, 5, 7, 9, 13–14, 19, and 27 belonged to plus-C (Table 2). A phylogenetic tree of 178 OBPs of *G. cantor* and 10 other species was constructed, and the results showed that many GcanOBPs were highly differentiated in different clades. Notably, GcanOBP3 and AglaOBP16, GcanOBP7 and AglaOBP15, GcanOBP23 and AglaOBP2, GcanOBP26 and AglaOBP11, and GcanOBP27 and MaltOBP3 were clustered together with a high degree of homology (Figure 2). 

### 3.3. Identification of Candidate CSPs

A total of 14 CSPs were identified in *G. cantor*. Their lengths ranged from 219 bp to 882 bp. Seven GcanCSPs had intact ORFs and four conserved cysteine sites, which are characteristic of typical insect CSPs. The homology searching results showed that 90.36% of GcanCSP4 were orthologs of the proteins in *C. bowringi* chemosensory protein 11, and 89.66% of GcanCSP7 were orthologs in *M. alternatus* chemosensory protein 2 (Table 2). A phylogenetic tree was constructed using 14 CSP sequences of *G. cantor* and 87 CSP sequences of 10 other species. Phylogenetic tree analysis showed that GcanCSP4, 5, 7 and 9 had high homology and close clustering with *M. alternatus* (Figure 3).

### 3.4. Identification of Candidate ORs

We identified and annotated 13 ORs, and GcanOR4, 6–10 had 7 transmembrane domains (TMDs) (Appendix A). GcanOR2 showed best matches with *A. germari* QNH68049.1 (97.15%). ORs from *G. cantor*, *A. germari*, *E. scrobiculatus*, *H. parallela*, *O. nubilalis*, *P. maculicollis* and *T. castaneum* were used to construct a neighbor-joining tree (Appendix A). In addition, GcanOR2 was on the same branch and highly homologous with OR co-receptor (ORco) of *A. glabripennis* and *C. bowringi*. In addition, GcanOR5 was clustered with McarOR20.

### 3.5. Identification of Candidate IRs

We identified 18 candidate IRs in *G. cantor*. All of the GcanIRs had complete ORFs encoded from 924 to 2877 amino acid residues, and GcanIR4 showed best matches with *A. glabripennis* ionotropic receptor 25a (92.95%) (Appendix A). The other 17 GcanIRs had 2 to 5 TMDs except for GcanIR1. Some GcanIRs were clustered together with high homology in the phylogenetic tree, including GcanIR1, GcanIR7, and GcanIR10 (Appendix A).

### 3.6. Identification of Candidate SNMPs

Two SNMPs genes were identified and both of them had complete ORFs with 451 aa and 515 aa (Appendix A). GcanSNMP1 showed matches with *M. alternatus* SNMP1 (80.89%). In the phylogenetic tree, GcanSNMP2 shared 100% identity with AglaSNMP2 (Appendix A).

### 3.7. Transcriptome Validation via Quantitative RT-PCR

The expression of OBP22, OBP25 and CSP4 was significantly upregulated in the female antennae of *G. cantor*, which was consistent with the RNA-Seq results (Figure 4). The data of the other selected unigenes were also consistent with the qRT-PCR results, demonstrating the reliability of our data and analysis (Figure 4).

### 3.8. Expression Patterns of OBP and CSP Genes

We measured the relative expression levels of OBP22, OBP25, and CSP4 at different ages of the adult antennae and different tissues of 30 d adult *G. cantor* by qRT-PCR to understand the function of OBPs and CSPs. The results (Figure 5) showed that the expression levels of GcanOBP22 and GcanOBP25 in females increased with age, but no significant difference was observed between the sexes. Furthermore, the expressions of GcanOBP22 and GcanOBP25 on the legs and wings were not significantly different from that of the antennae. GcanCSP4 was significantly highly expressed in the antennae, and the highest expression in the female antennae occurred at 12 d, which was also significantly higher than that of males.

## 4. Discussion

With the continuous development of sequencing technology, transcriptome analysis has become increasingly widely used in chemically accepted gene identification of different coleoptera insects [34,35]. In this context, insect olfactory genes have been identified extensively because they play an important role in the life history of insects. *G. cantor*, as a species of beetle, is a pest that is extremely harmful to urban ecology and economy. However, information on the olfactory mechanism of *G. cantor* is limited. In this study, we established the antennal transcriptomes and obtained 76 candidate olfactory genes to probe into their crucial role in *G. cantor*, including 29 OBPs, 14 CSPs, 13 ORs, 18 IRs, and 2 SNMPs. The results of this study not only provide new ideas and methods for further research of the chemosensory system of beetles, but also lays a molecular foundation for the development of novel attractants and repellants.

OBPs play an important role in the first step of odor detection by identifying, screening, binding, and transporting molecules [35]. We identified 29 OBPs in the antennal transcriptome of *G. cantor*, which was fewer than that in *T. castaneum* (49 OBPs) [36], *A. glabripennis* (42 OBPs) [37], and *M. alternatus* (52 OBPs) [38]. In the phylogenetic tree, GcanOBP3, GcanOBP7, GcanOBP20, and GcanOBP26 clustered in the same clade with *A. glabripennis*, which indicated that they might have a close kinship and similar functions [39]. GcanOBP22 and GcanOBP25 displayed higher expression in female antennae of 30 d adults among different tissues, but the antennae were not significantly different between sexes, which might play a role in identifying host plant volatiles [40,41]. Furthermore, the expression of GcanOBP22 and GcanOBP25 on the legs and wings was not significantly different from that on the antennae. In addition, the expression of GcanOBP22 and GcanOBP25 on the male antennae and legs was not significantly different, which had shown that the high expression of OBP on male antennae might play a role in recognizing female sex pheromone [42]. Whether the high expression on the legs has this function requires further experiments. Research had shown that high expression on insect legs modulates behavioral adaptations to host plants [43,44]. OBPs were expressed in non-olfactory tissues, so it was considered that they might serve as carriers of chemicals during developmental and physiological processes [44,45].

In the chemical communication of insects, CSPs are known as secondary classes of OBPs [46,47]. The cystic structure of CSPs is the carrier of hydrophobic substances, which play a key role in the transmission of pheromones, transduction of taste, dissolution, and absorption of nutrients, growth, and development of insects, and drug resistance [13,48]. A total of 14 CSPs were found in *G. cantor*, which was more than that in *A. glabripennis* (12) but fewer than that in *H. parallela* (16), *M. alternatus* (19), and *T. castaneum* (20) [36,37,38]. The results indicated that CSPs in coleopteran insects have duplication and differentiation under the role of natural selection [49]. Moreover, we found that GcanCSP4 was significantly highly expressed and female-biased in the antennae of 30 d adults, which might be related to the recognition of pheromones and plant volatiles [50,51]. Also, it was the highest expression on the female antennae at 12 d, which might be one of the genes that binds to the oviposition marking pheromone when the female lays eggs [35,51,52]. Based on the expression pattern of GcanCSP4, it might be a gene closely related to female egg-laying behavior.

ORs are important chemoreceptors, which primarily detect sex pheromones and odorants [53]. We identified 13 ORs in *G. cantor*, which was much fewer than those in *A. glabripennis* (37 ORs), *A. corpulenta* (43 ORs), *D. ponderosae* (49 ORs), and *T. castaneum* (111 ORs) [36,37,54,55], but was similar to those of *M. alternatus* (9 ORs) [38]. Some ORs might have been missed in our transcriptome analysis and caused the true number of ORs expressed in the antenna in this beetle to be underestimated. Recent studies have suggested that the number of ORs might be influenced by physiologically dependent gene expression levels [56]. In addition, it might be the reality that the number of ORs present in a small number because *G. cantor* is oligophagous on kapok [23], which can be targeted easily to our identified behaviorally active kapok volatile (unpublished data) with the help of the GcanORco, and needs further confirmation. GcanOR2 was on the same branch and highly homologous with ORco of other species, so GcanOR2 might be the ORco gene of *G. cantor* that ORco is a highly conserved receptor found in various insects [35]. GcanOR5 was clustered with pheromone receptor (McarOR20), which indicated that GcanOR5 might be PR of *G. cantor* [57].

IRs are sensory proteins that evolved from ionotropic glutamate receptors (iGluRs). They are primarily found in taste organs/receptors that respond to food components such as sugar, salt, water, and bitter compounds and detect minute temperature differences [18,58]. Usually, IRs are expressed in combinations of sensory neurons that respond to many different odors [35,59]. A total of 18 IRs were identified in *G. cantor*, which was more than those of *A. glabripennis* (4) [37].

SNMPs are the transmembrane domain proteins, which play different crucial roles in the peripheral olfactory system [60]. Most studies have shown two types of SNMP genes in insects, namely SNMP1 and SNMP2 [47]. Two SNMPs were also identified in this study. A previous study found that SNMP1 was primarily expressed in *M. mediator* antennae, which might be related to the perception of plant volatiles and sex pheromones [61], indicating that GcanSNMP1 might also have a similar effect.

In this study, there are some points that should be addressed. First, a total of 7019 unigenes were classified into human disease categories which seems to be unexplainable. Some insect antennal transcriptome analysis also pointed out that human diseases were one of the numerically dominant categories among all the annotated unigenes into KEGG, such as *Heortia vitessoides* and *Zele chlorophthalmus* [62,63], which might participate in the defense, hormone, antioxidative and other metabolic processes in insects and play a role in human disease research in the long run. Second, we used qRT-PCR to measure the expression level of some interesting genes, while immunohistochemistry is a useful way to locate them in the cells. This could be studied in the future with a combined analysis with our previous work on antennal sensilla [28].

## 5. Conclusions

In this study, a total of 76 olfactory-related genes were annotated and identified, including 29 OBPs, 14 CSPs, 13 ORs, 18 IRs, and 2 SNMPs. Furthermore, the differential expressions of OBP22, OBP25, CSP4, IR6, IR8, IR9, IR10, and IR18 in the antennae were analyzed by qRT-PCR, which was consistent with the results of RNA-seq. Furthermore, the expression patterns of GcanOBP22, GcanOBP25, and GcanCSP4 were antennae biased. GcanOBP22 and GcanOBP25 might play a role in identifying host plant volatiles. In addition, GcanCSP4 might be a gene closely related to female egg-laying behavior. The results of this study provided a theoretical basis for studying the combination and perception mechanism of beetle’s olfactory genes and ligands and new insights into the development of efficient olfactory attractants.

## Figures and Tables

**Figure 1 insects-13-00553-f001:**
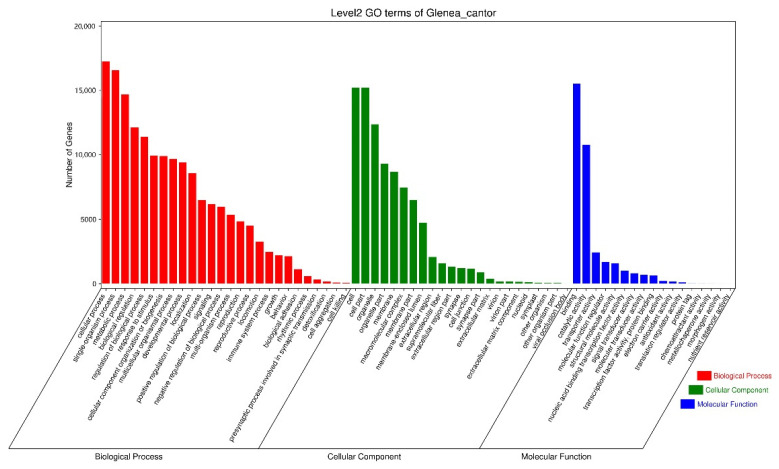
GO Analysis of *Glenea cantor* antennal transcriptome.

**Figure 2 insects-13-00553-f002:**
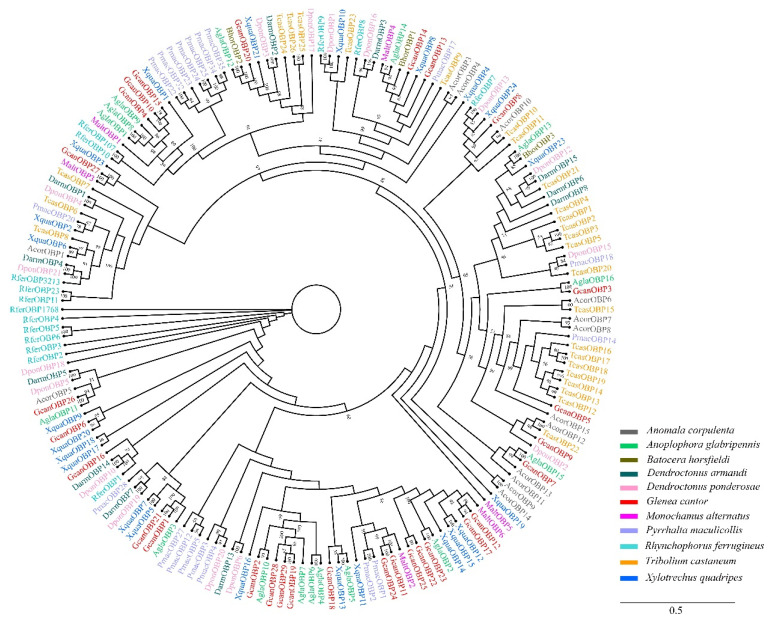
Phylogenetic analysis of insect odorant binding proteins (OBPs). Information of OBPs was listed in Appendix A. *Anomala corpulenta* (Acor), *Anoplophora glabripennis* (Agla), *Batocera horsfieldi* (Bhor), *Dendroctonus armandi* (Darm), *Dendroctonus ponderosae* (Dpon), *Monochamus alternatus* (Malt), *Pyrrhalta maculicollis* (Pmac), *Rhynchophorus ferrugineus* (Rfer), *Tribolium castaneum* (Tcas), and *Xylotrechus quadripes* (Xqua).

**Figure 3 insects-13-00553-f003:**
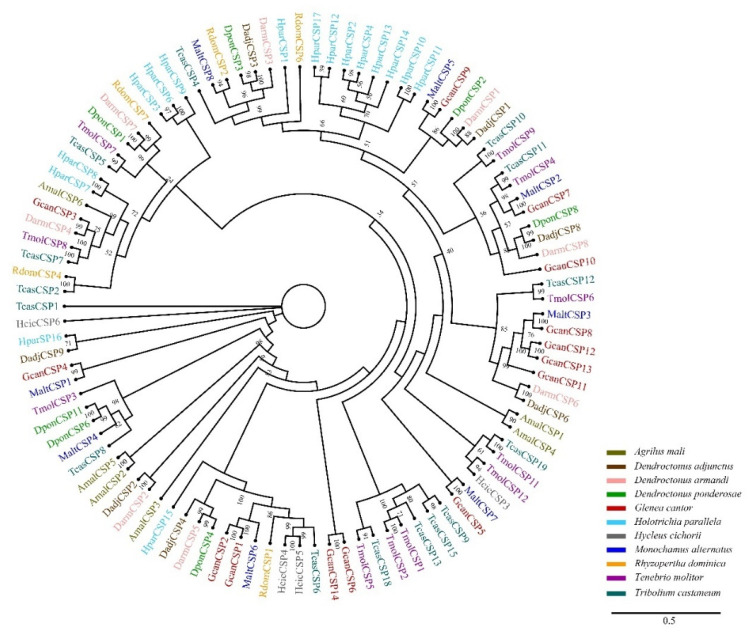
Phylogenetic analysis of insect chemosensory proteins (CSPs). Information of CSPs was listed in Appendix A. *Agrilus mali* (Amal), *Dendroctonus adjunctus* (Dadj), *Dendroctonus armandi* (Darm), *Dendroctonus ponderosae* (Dpon), *Holotrichia parallela* (Hpar), *Hycleus cichorii* (Hcic), *Monochamus alternatus* (Malt), *Rhyzopertha dominica* (Rdom), *Tenebrio molitor* (Tmol), and *Tribolium castaneum* (Tcas).

**Figure 4 insects-13-00553-f004:**
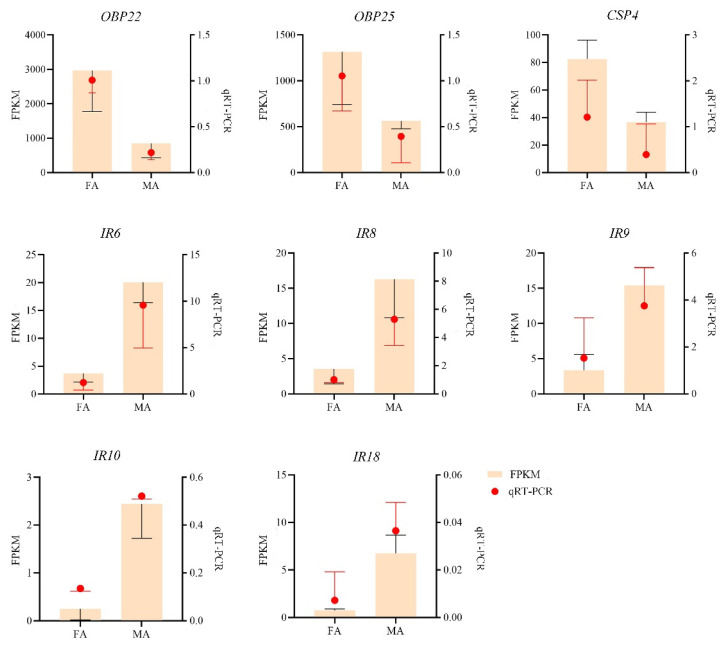
Validation of gene expression by qRT-PCR of selected genes. These genes included *OBP22*, *OBP25*, *CSP4*, *IR6*, *IR8*, *IR9*, *IR10*, and *IR18* of *Glenea cantor*. Orange bars indicate the FPKM values (*y*-axis on left) and the red dots represent the relative expression level (*y*-axis on right). FA: female antennae; MA: male antennae. Data were shown as mean ± SD (*n* =3).

**Figure 5 insects-13-00553-f005:**
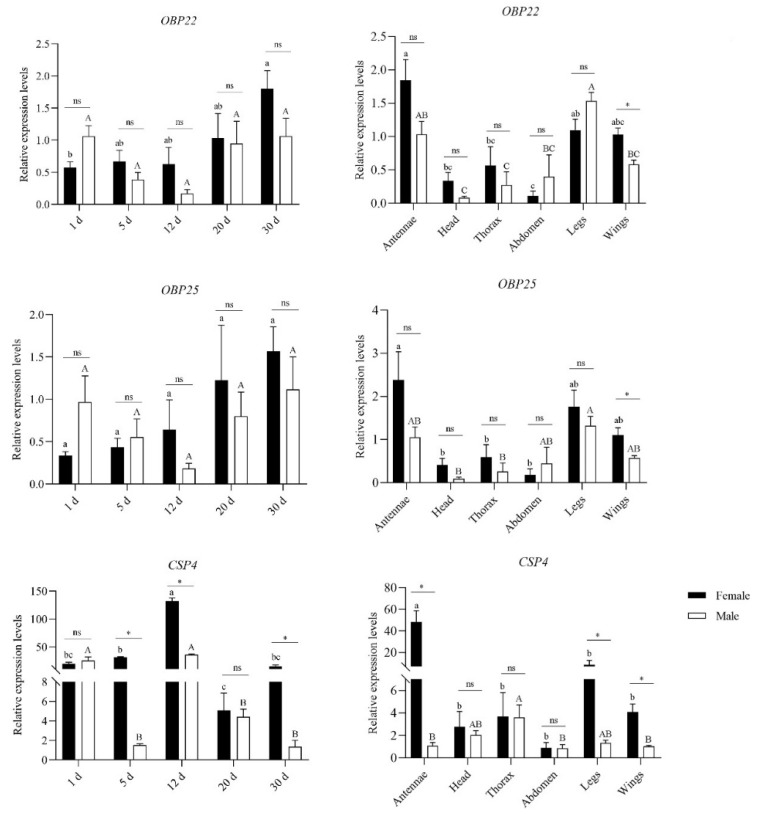
Relative expression levels of *OBP22*, *OBP25*, and *CSP4* of adult *Glenea cantor* antennae at different ages and of different tissues of 30 d adult. The expression levels were calculated using the 2^−ΔΔCt^ method and the male antennae (30 d) were selected as the calibrator to normalize the gene expression levels in various ages and tissues. The values were shown as means ± SE (*n* = 3). Different lowercase letters indicate significant differences between ages or tissues of females and different capital letters indicate significant differences between ages or tissues of males (*p* < 0.05, ANOVA). * indicates significant difference between different sexes in the same age or tissue. ns indicates no significant difference.

**Table 1 insects-13-00553-t001:** Statistics of the antennal transcriptome of *Glenea cantor*.

	Female	Male
Raw reads	40,989,086	42,482,724
Clean reads	38,130,472	42,395,862
N percentage	0.00%	0.00%
Q20 percentage	99.78%	99.80%
GC percentage	48.82%	48.42%
Assembly unigene	24,462	
GC percentage	44.18%	
N50	8089	
N50 max length (bp)	12,058	
N50 min length (bp)	116	

**Table 2 insects-13-00553-t002:** OBPs and CSPs identified in *Glenea cantor*.

Gene Name	Unigene ID	Unigene Length (bp)	ORF (aa)	Complete ORF	Signal Peptide	Cysteine Number	Homology Search with Known Protein				
							Name	Species	E-Value	Accession	Identity (%)
OBP1	Isoform0013634	189	62	YES	0	2	odorant binding protein 7	*Xylotrechus quadripes*	8 × 10^−20^	AXO78385.1	62.9
OBP2	Isoform0013768	351	116	YES	0	3	odorant binding protein 7	*Anoplophora glabripennis*	3 × 10^−20^	ARH65462.1	50
OBP3	Isoform0018207	534	177	YES	22	10	odorant binding protein	*Anoplophora chinensis*	2 × 10^−99^	AUF72969.1	80.34
OBP4	Isoform0021420	300	99	YES	20	4	odorant binding protein 1	*Monochamus alternatus*	5 × 10^−22^	ABR53888.1	55.68
OBP5	Isoform0022365	702	233	YES	26	11	minus-C odorant binding protein 4, partial	*Anoplophora glabripennis*	6 × 10^−76^	ATP75519.1	79.5
OBP6	Isoform0022758	384	127	YES	20	5	odorant-binding protein 18	*Monochamus alternatus*	3 × 10^−18^	AIX97033.1	40.5
OBP7	Isoform0023655	744	247	YES	17	13	odorant-binding protein 24	*Monochamus alternatus*	2 × 10^−82^	AIX97039.1	73.53
OBP8	Isoform0023835	474	157	YES	21	6	odorant-binding protein 14	*Monochamus alternatus*	5 × 10^−63^	AIX97029.1	79.49
OBP9	Isoform0023942	669	203	YES	19	10	odorant-binding protein 19	*Monochamus alternatus*	5 × 10^−86^	AIX97034.1	70.39
OBP10	Isoform0024133	432	143	YES	20	5	odorant-binding protein 1	*Monochamus alternatus*	1 × 10^−37^	ABR53888.1	50.36
OBP11	Isoform0024165	405	134	YES	18	5	odorant-binding protein 2	*Monochamus alternatus*	2 × 10^−64^	AHA39267.1	78.51
OBP12	Isoform0024173	429	142	YES	17	5	odorant-binding protein 6	*Monochamus alternatus*	3 × 10^−66^	AJO67868.1	73.33
OBP13	Isoform0024208	429	142	YES	20	8	odorant-binding protein 4	*Monochamus alternatus*	1 × 10^−69^	AHA39269.1	69.01
OBP14	Isoform0024216	429	142	YES	20	9	odorant binding protein 14	*Anoplophora glabripennis*	2 × 10^−71^	ARH65469.1	80.31
OBP15	Isoform0024236	432	143	YES	20	5	odorant-binding protein 1	*Monochamus alternatus*	2 × 10^−39^	ABR53888.1	51.09
OBP16	Isoform0024238	399	132	YES	18	4	odorant-binding protein 20	*Monochamus alternatus*	2 × 10^−41^	AIX97035.1	52.27
OBP17	Isoform0024250	405	134	YES	17	6	odorant binding protein 6	*Monochamus alternatus*	5 × 10^−73^	AJO67868.1	81.48
OBP18	Isoform0024256	402	133	YES	17	4	odorant binding protein 5	*Anoplophora glabripennis*	9 × 10^−28^	ARH65460.1	34.65
OBP19	Isoform0024263	420	139	YES	22	7	odorant binding protein 7	*Anoplophora glabripennis*	5 × 10^−44^	ARH65462.1	56.43
OBP20	Isoform0024283	414	137	YES	19	6	odorant-binding protein 10	*Monochamus alternatus*	2 × 10^−81^	AIX97025.1	83.21
OBP21	Isoform0024286	405	134	YES	18	5	odorant binding protein 7	*Xylotrechus quadripes*	3 × 10^−520^	AXO78385.1	60.61
OBP22	Isoform0024293	402	133	YES	17	5	odorant-binding protein 5	*Monochamus alternatus*	3 × 10^−82^	AIX97020.1	89.47
OBP23	Isoform0024294	429	142	YES	25	5	odorant-binding protein 11	*Monochamus alternatus*	5 × 10^−75^	AIX97026.1	82.48
OBP24	Isoform0024314	405	134	YES	18	5	odorant-binding protein 2	*Monochamus alternatus*	2 × 10^−64^	AHA39267.1	78.51
OBP25	Isoform0024319	405	134	YES	17	4	odorant-binding protein 5	*Monochamus alternatus*	6 × 10^−73^	AIX97020.1	85.07
OBP26	Isoform0024320	390	129	YES	17	4	odorant-binding protein 21	*Monochamus alternatus*	4 × 10^−41^	AIX97036.1	55.04
OBP27	Isoform0024323	462	136	YES	21	7	odorant-binding protein 3	*Anoplophora glabripennis*	1 × 10^−59^	ATO58974.1	61.03
OBP28	Isoform0024327	423	140	YES	20	5	odorant binding protein 7	*Anoplophora glabripennis*	5 × 10^−41^	ARH65462.1	52.71
OBP29	Isoform0024357	420	139	YES	23	6	odorant binding protein 7	*Anoplophora glabripennis*	3 × 10^−43^	ARH65462.1	55.22
CSP1	Isoform0017649	882	293	YES	21	4	chemosensory protein 6	*Monochamus alternatus*	5 × 10^−74^	AIX97046.1	82.58
CSP2	Isoform0019553	840	279	YES	21	4	chemosensory protein 6	*Monochamus alternatus*	3 × 10^−74^	AIX97046.1	82.58
CSP3	Isoform0023484	375	124	YES	20	4	chemosensory protein CSP8	*Tenebrio molitor*	6 × 10^−54^	AJO62214.1	76.92
CSP4	Isoform0023726	357	118	YES	24	5	chemosensory protein 11	*Colaphellus bowringi*	2 × 10^−49^	ALR72525.1	90.36
CSP5	Isoform0024206	378	125	YES	23	4	chemosensory protein 7	*Monochamus alternatus*	1 × 10^−46^	AIX97047.1	63.79
CSP6	Isoform0024227	378	125	YES	18	5	CSP9	*Anoplophora glabripennis*	4 × 10^−53^	ATL75742.1	61.6
CSP7	Isoform0024262	219	72	YES	22	0	chemosensory protein 2	*Monochamus alternatus*	1 × 10^−10^	AIX97042.1	89.66
CSP8	Isoform0024264	381	126	YES	19	4	chemosensory protein 11	*Monochamus alternatus*	1 × 10^−59^	AIX97086.1	77.06
CSP9	Isoform0024328	390	129	YES	19	5	chemosensory protein 5	*Monochamus alternatus*	1 × 10^−59^	AIX97045.1	80.91
CSP10	Isoform0024344	387	128	YES	19	5	chemosensory protein 8	*Dendroctonus ponderosae*	5 × 10^−50^	AGI05164.1	61.79
CSP11	Isoform0024345c	381	126	YES	19	5	chemosensory protein 11	*Monochamus alternatus*	1 × 10^−69^	AIX97086.1	75.2
CSP12	Isoform0024347	381	126	YES	19	4	chemosensory protein 11	*Monochamus alternatus*	2 × 10^−75^	AIX97086.1	85.6
CSP13	Isoform0024362	381	126	YES	19	4	chemosensory protein 11	*Monochamus alternatus*	5 × 10^−66^	AIX97086.1	82.4
CSP14	Isoform0024366	378	125	YES	18	5	CSP9	*Anoplophora glabripennis*	4 × 10^−53^	ATL75742.1	61.6

## Data Availability

The raw data that support the findings of this study are available from NCBI database of antennae transcriptomes submission number SRR17331115-SRR17331120.

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
