# Peer review of "Antennal Transcriptome Analysis and Identification of Olfactory Genes in Glenea cantor Fabricius (Cerambycidae: Lamiinae)"

_insects, 2022, doi:10.3390/insects13060553_

Round 1
Reviewer 1 Report
This paper reports an analysis of antennal transcriptome from Glenea cantor Fabricius (Cerambicidae: Lamiinae), one of the pests of the kapok tree. To look for the genes which can be served for molecular based pesticide, the authors conducted whole antennae transcriptome analysis and then focuses on olfactory genes, i.e., odorant binding proteins (OBPs), chemosensory proteins (CSPs), odorant receptors (ORs), ionotropic receptors (IRs), and sensory neuron membrane proteins (SNMPs). As a result, the study identified 76 olfactory-related genes. In addition, the author conducted transcriptomic analyses of three genes of different age, different tissues, and different sexes. This could bethe largest transcriptomic data on G. cantor that has been published so far. To a non-model and poorly studied organism, this paper provides a starting point for the understanding of olfactory mechanism of the species, which mayhelp to find a more efficient way to control this pest. However, I found this manuscript is pretty dry, likely due to the purpose or rationale of each analysis is not clear stated.
Major comments
1. The authors need to state how they preprocessed raw reads before de novo analysis, such as adaptor control, base quality control, length and so on, in the Materials and Methods or Results. In addition, it will be better to specify parameters and threshold that were set in the section.
2. the ratio of identified ORs and IRs (13:18) is much smaller than that of other insects (e.g., S. noctilio, S. nitobe, D. melanogaster, and An. gambiae). In addition, a small subset of gustatory receptors (GRs) was uncovered in the antennae of insects mentioned above but no GR was identified here. Do authors have any thought?
3. Following above comment, the authors explained that some ORs might have been missed in their analyses (Discussion, line 330-332). If so, what is the coverage of genes expressed in the antennae of G. cantor in this work?
4. Olfactory receptor genes are subjected to different fates during evolution, such as gene gain and loss (Nozawa et al. (2007) PNAS 104(17):7122-7127; Missbach et al. (2014) eLife 2014(3):1-22; Ramasmy et al. (2016) Genome Biol. Evol. 8(8):2297-2311; Sanchez-Garcia et al. (2009) Heredity 103(3):208-216.). Thus, using de novo assembly may lead to chimeric sequences between closely related genes. How did the authors avoid/address the problems?
5. (Line 176) The authors mentioned that N50 was 8089 from 116 bp to 12058 bp. Did they do any statistical analysis to show the quality of this assembled transcriptome?
6. It is not clear whether the authors did GO analysis for all assembled genes or just a subset of these. What are 289,811 genes in GO analysis (line 180), 28,541 genes in KEGG analysis (line 182), and 24,448 genes in KOG analysis (line 188)?
7. (Line 194-195) The authors observed the 10 olfactory genes were sexual-biased. It will be better to explain how these genes were identified (under what kind of comparison and criteria), what the sexual-biased expressions are (e.g., increased or decreased, etc.), and refer them to Fig S2.
8. It will be better to show the phylogenetic tree of all species described in Fig 2, Fig S3, Fig S4 and Fig S5. It is required to explain why genes from different numbers of species were subjected to phylogenetic analyses in these figures. It is also better that the authors address the problem of chimeric sequences (as mentioned in comment #3) to show that the inputs are reliable. In addition, what the point of building phylogenetic tree is also needed to be clarified.
9. Figure legends are too short and do not describe how to read the figure. Figure 4 is totally unreadable.
10. It is lacking of rationale why only the expressions of OBP and CSP genes were subjected for differential age and tissue analysis (Fig 5).
11. (Fig. 5) Why did the authors choose these ages? Whether it is related to the reproductive stage or special behaviors of the beetles? Why OBP22, OBP23 and CSP4 are expressed in some tissues other than antennae? In addition, the age-dependent changes of these three genes showed different tends in females and males and across these three genes. This is interesting but obscure that some OBPs and CSPs (but not any ORs and IRs) show sexual-biased and age-dependent expression patterns. In general, the binding specificity of odorants is highly relied on ORs and IRs. The authors may need to discuss this.
12. ANOVA test was used for the analyses shown in Fig 5. A post hoc correction may be needed.
13. (Line 347-348) The authors cite reference #28 to claim that taste receptors were widely distributed on G. cantor antennae. This paper (ref #28) did not touch anything about taste receptors. Therefore, it is not clear how they could propose that GcanIRs (in the antenna) might sever as taste receptors. This is a big gap.
14. (Line 369-370) The authors stated that GcanOBP22, GcanOBP25 and GcanCSP4 are antennae biased. However, they didn’t compare the expressions of ORs or IRs in different tissues. Therefore, the following conclusion that “...indicating that they might play an important role in olfactory perception” is meaningless. On the other hand, the authors may discuss why other tissues, especially legs and wings, express these three genes in relatively high levels than other non-antennae tissues. (Also see comment #6.)
15. Many sequences that the author used for their analysis are not available (provided in the supplementary). The authors should deposit to NCBI so that they are accessible to others and have IDs.
Minor comments
1. (line 44) “......., which can dissolve volatile molecules, bind and transfer such molecules to the receptors, ...” reads weird. Can OBPs dissolve volatile molecules?
2. In the Materials and Methods section, it will be better to mention whether RNAs of other tissues than antennae were also extracted in the same way as they did for antennae.
3. (Line 103-104) It is confusing why the authors wanted to enrich prokaryotic mRNA.
4. It will be better to specify the version of the databases (or accessed date). In addition, it will be better to clarify BLAST parameters used to filter the best match (i.e., query coverage, identity, e-value, etc.) (Also see major comment #2.)
5. (Line 127) The authors claimed that they downloaded coleopteran sequences, but at least one species listed does not belong coleopteran (Bombyx mori).
6. Whether the authors used the same sequenced RNA samples or prepared independent replicates? This needs to be clarified.
7. (Fig 1) The font size of most annotations in this figure is too small and barely recognized.
8. (Line 202) Why are the first few words bolded?
9. (Table 2) The number of cysteine is less meaningful. The authors may show if their sequences contain the right motif.
10. One species was missed in the figure legend of Fig S4.
11. Although both qRT-PCR and RT-qPCR are accepted as abbreviation for quantitative real-time PCR, the authors should choose either one to use throughout the paper.
12. (Line 110) “…DGEs analysis” should be “…DGE analysis”.
Author Response
Dear Reviewer 1,
We really appreciate your earnest and careful review of our manuscript ID: insects-1763077 entitled as “Antennal transcriptome analysis and identification of olfactory genes in Glenea cantor Fabricius (Cerambycidae: Lamiinae)”. We have revised the manuscript accordingly. Our point-by-point responses are detailed below.
Point 1: The authors need to state how they preprocessed raw reads before de novo analysis, such as adaptor control, base quality control, length and so on, in the Materials and Methods or Results. In addition, it will be better to specify parameters and threshold that were set in the section.
Response 1: It has been added. ”Reads obtained from the sequencing machines included raw reads containing adapters or low quality bases which would affect the following assembly and analysis. Thus, to get high quality clean reads, reads were further filtered by fastp (version 0.18.0). The parameters were as follows: (1) removing reads containing adapters; (2) removing reads containing more than 10% of unknown nucleotides (N); (3) removing low quality reads containing more than 50% of low quality (Q-value≤20) bases.”
Point 2: the ratio of identified ORs and IRs (13:18) is much smaller than that of other insects (e.g., S. noctilio, S. nitobe, D. melanogaster, and An. gambiae). In addition, a small subset of gustatory receptors (GRs) was uncovered in the antennae of insects mentioned above but no GR was identified here. Do authors have any thought?
Response 2: First, the specificity of species and might be an explanation, such as Monochamus alternatus (OR: IR = 10: 8), Anoplophora glabripennis (37: 4). Second, most GRs are expressed in gustatory receptor neurons in taste organs and are involved in contact chemoreception. These GRs typically detect different sugars, bitter compounds, and contact pheromones (Vosshall and Stocker, 2007). In this work, olfactory-related genes were mainly analyzed, so taste receptors were not considered.
Point 3: Following above comment, the authors explained that some ORs might have been missed in their analyses (Discussion, line 330-332). If so, what is the coverage of genes expressed in the antennae of G. cantor in this work?
Response 3: Compared with some beetles e.g., A. glabripennis (37 ORs), A. corpulenta (43 ORs), D. ponderosae (49 ORs), and T. castaneum (111 ORs), which were generally more plentiful in number than G. cantor. Therefore, it was speculated that there might be missed. Since transcriptome analysis is not good enough to uncover novel genes, there might be possible ORs which are specific in G. cantor. Gene coverage is the percentage of the genes covered by the mapped reads, calculated as the ratio of the number of covered bases to the number of total bases in the coding regions. In this work, the coverage of genes expressed in the antennae of G. cantor were 99.77%-99.81%.
Point 4: Olfactory receptor genes are subjected to different fates during evolution, such as gene gain and loss (Nozawa et al. (2007) PNAS 104(17):7122-7127; Missbach et al. (2014) eLife 2014(3):1-22; Ramasmy et al. (2016) Genome Biol. Evol. 8(8):2297-2311; Sanchez-Garcia et al. (2009) Heredity 103(3):208-216.). Thus, using de novo assembly may lead to chimeric sequences between closely related genes. How did the authors avoid/address the problems?
Response 4: It is ture that de novo assembly might lead to chimeric sequences as you referred. In fact, it is a common problem that the quality of de novo assembled transcriptomes is difficult to assess. Besides N50, BUSCO was used to assess assembly quality.
Point 5: (Line 176) The authors mentioned that N50 was 8089 from 116 bp to 12058 bp. Did they do any statistical analysis to show the quality of this assembled transcriptome?
Response 5: Quality assessment of assembly results can be assessed from the N50 value. All Unigene is sorted from long to short, and the length is accumulated in turn. When the accumulated fragment length reaches 50% of the total fragment length (the length of all Unigenes), the length and quantity of the corresponding fragment are the length and quantity of Unigene N50. The longer the Unigene N50, the smaller the quantity, the better the assembly quality.
Point 6: It is not clear whether the authors did GO analysis for all assembled genes or just a subset of these. What are 289,811 genes in GO analysis (line 180), 28,541 genes in KEGG analysis (line 182), and 24,448 genes in KOG analysis (line 188)?
Response 6: GO analysis was performed on all assembled genes. “Gene Ontology (GO) annotation analysis showed that 289,811 unigenes could be annotated into three functional categories.”, “KEGG pathway annotation analysis showed that 28,541 genes could be annotated into 6 KEGG categories”, “COG/KOG analysis showed that a total of 24,448 unigene sequences were classified into 25 COG functional categories.”
Point 7: (Line 194-195) The authors observed the 10 olfactory genes were sexual-biased. It will be better to explain how these genes were identified (under what kind of comparison and criteria), what the sexual-biased expressions are (e.g., increased or decreased, etc.), and refer them to Fig S2.
Response 7: The input data for gene differential expression analysis were reads count data obtained from gene expression level analysis, which was analyzed by DESeq2 software (Subramanian et al., 2005). The analysis was mainly divided into three parts: (1) Normalization of readcount; (2) Calculate the probability of hypothesis testing (pvalue) according to the model; (3) Finally, multiple hypothesis testing is performed to obtain the FDR value (error detection rate). Based on the results of variance analysis we screening the genes of FDR < 0.05 and |log2FC| > 1 for significant differences. ”By comparing the FPKM values, it was found that there were 10 olfactory genes which were sex-biased from transcriptome. Among them, including OBP22, OBP25, CSP4, and CSP10 were female-biased, IR6, IR8, IR9, IR10, IR15, and IR18 were male-biased.”
Point 8: It will be better to show the phylogenetic tree of all species described in Fig 2, Fig S3, Fig S4 and Fig S5. It is required to explain why genes from different numbers of species were subjected to phylogenetic analyses in these figures. It is also better that the authors address the problem of chimeric sequences (as mentioned in comment #3) to show that the inputs are reliable. In addition, what the point of building phylogenetic tree is also needed to be clarified.
Response 8: Long-horned beetle is a coleopteran pest. Therefore, all the reference species fall into this category. Since it’s unrealistic to cover all of the coleopteran species, the BLASTX by NCBI were used to select species which showed a certain homology with G. cantor during the olfactory gene analysis. Some other references selected typical species from different insect order as reference for evolutionary analysis which were not suitable for our objective in this study.Our objective was not only to identify olfactory genes but also to provide priority target genes for possible functional analysis. Therefore, BLASTX was used as introduced above to focus reference species, and some species only report limited olfactory genes which belonged to some kinds of them. As a result, olfactory gene categories with more published genetic information and functional analysis were able to involve more species, i.e. OBP and CSP. Transcriptome quality control is an important step in RNA-Seq experiments. Removing chimeric sequences could use a novel method named “Bellerophon” (Kerkvliet et al., 2019). It first uses the quality assessment tool TransRate to indicate the quality, after which it uses a transcripts per million filter to remove lowly expressed contigs and remove highly identical contigs. building phylogenetic tree is to analyze homology with coleoptera and to better analyze the function of genes.
Point 9: Figure legends are too short and do not describe how to read the figure. Figure 4 is totally unreadable.
Response 9: Thank you for your suggestion. Figure 4 legends has been changed to “Validation of gene expression by qRT-PCR of selected genes.These genes included OBP22, OBP25, CSP4, IR6, IR8, IR9, IR10, and IR18 of Glenea cantor. Orange bars indicate the FPKM values (y-axis on left) and the red dots represent the relative expression level (y-axis on right). FA: female antennae; MA: male antennae. Data were shown as mean ± SD (n =3).
Point 10: It is lacking of rationale why only the expressions of OBP and CSP genes were subjected for differential age and tissue analysis (Fig 5).
Response 10: OBPs and CSPs play an important role in the first step of odor detection. In addtion, sexual differentially expressed of them might be involeved in mating and reproduction which is gender-associated. In follow-up experiments we would like to continue to explore their functions
Point 11: (Fig. 5) Why did the authors choose these ages? Whether it is related to the reproductive stage or special behaviors of the beetles? Why OBP22, OBP23 and CSP4 are expressed in some tissues other than antennae? In addition, the age-dependent changes of these three genes showed different tends in females and males and across these three genes. This is interesting but obscure that some OBPs and CSPs (but not any ORs and IRs) show sexual-biased and age-dependent expression patterns. In general, the binding specificity of odorants is highly relied on ORs and IRs. The authors may need to discuss this.
Response 11: Firstly, the adult G. cantor have gone through mating and egg-laying at 30-day old, and both female and male are mature. Secondly, age is a factor that influence gene expression. In this study, we also would like to probe into the sexual different expression in antenna. Therefore, the antenna from 30-day female and male were applied into comparative transcriptome analysis. We agree with you that “the binding specificity of odorants is highly relied on ORs and IRs”, We also found a possible ORco gene of G. cantor through homology comparison of phylogenetic tree, but OBPs and CSPs are the first step of odor detection. In this paper we tend to explore the expression patterns of OBPs and CSPs, and in follow-up experiments we will focus on OR and IR genes. a
Point 12: ANOVA test was used for the analyses shown in Fig 5. A post hoc correction may be needed.
Response 12: Before ANOVA test, test data conforming to normal distribution.
Point 13: (Line 347-348) The authors cite reference #28 to claim that taste receptors were widely distributed on G. cantor antennae. This paper (ref #28) did not touch anything about taste receptors. Therefore, it is not clear how they could propose that GcanIRs (in the antenna) might sever as taste receptors. This is a big gap.
Response 13: Thanks for four reminder. This sentence has been removed.
Point 14: (Line 369-370) The authors stated that GcanOBP22, GcanOBP25 and GcanCSP4 are antennae biased. However, they didn’t compare the expressions of ORs or IRs in different tissues. Therefore, the following conclusion that “...indicating that they might play an important role in olfactory perception” is meaningless. On the other hand, the authors may discuss why other tissues, especially legs and wings, express these three genes in relatively high levels than other non-antennae tissues. (Also see comment #6.)
Response 14: Thanks for your suggestion. “...indicating that they might play an important role in olfactory perception” has been removed. There was a discussion of high expression in non-antennae tissues. “Research had shown that high expression on insect legs modulates behavioral adaptations to host plants. OBPs were expressed in non-olfactory tissues, which was considered that they might serve as carriers of chemicals during developmental and physiological processes.”
Point 15: Many sequences that the author used for their analysis are not available (provided in the supplementary). The authors should deposit to NCBI so that they are accessible to others and have IDs.
Response 15: We have uploaded the sequence to NCBI and can get GenBank accession number (SRR17331115-SRR17331120) and open on June 6, 2022.
Point 16: (line 44) “......., which can dissolve volatile molecules, bind and transfer such molecules to the receptors, ...” reads weird. Can OBPs dissolve volatile molecules?
Response 16: The description of OBPs has been modified. “OBPs are a class of small soluble proteins widely present in the lymph fluid of the antennal sensilla, which can bind and transfer such molecules to the receptors.”
Point 17: In the Materials and Methods section, it will be better to mention whether RNAs of other tissues than antennae were also extracted in the same way as they did for antennae.
Response 17: It has been added. “Whether RNA from other tissues is also extracted in the same way as the antennae and the experiment procedure was the same as above.”
Point 18: (Line 103-104) It is confusing why the authors wanted to enrich prokaryotic mRNA.
Response 18: Thanks for your reminder. “After extraction of total RNA, eukaryotic mRNA was enriched by Oligo(dT) beads, while prokaryotic mRNA was enriched by removing rRNA by Ribo-ZeroTM Magnetic Kit (Epicentre, Madison, WI, USA).” has been changed to “Eukaryotic messenger RNA (mRNA) was enriched from the total RNA by Oligo (dT) beads.“
Point 19: It will be better to specify the version of the databases (or accessed date). In addition, it will be better to clarify BLAST parameters used to filter the best match (i.e., query coverage, identity, e-value, etc.) (Also see major comment #2.)
Response 19: We have uploaded the sequence to NCBI and access dates were already open on June 6, 2022. The BLAST parameters searches using BLASTX (https://blast.ncbi.nlm.nih.gov/, accessed in December 2021). “Unigenes were annotated by using the BLASTx project (http://www.ncbi.nlm.nih.gov/BLAST/) with an E-value threshold of 1E-5.”
Point 20: (Line 127) The authors claimed that they downloaded coleopteran sequences, but at least one species listed does not belong coleopteran (Bombyx mori).
Response 20: Thank you for your remind. ”Bombyx mori” has been removed and the phylogenetic trees were checked again. Other species have been verified to be coleopteran.
Point 21: Whether the authors used the same sequenced RNA samples or prepared independent replicates? This needs to be clarified.
Response 21: To validate the transcriptome, the same sequenced RNA samples were used.
Point 22: (Fig 1) The font size of most annotations in this figure is too small and barely recognized.
Response 22: The picture is too small to see clearly, so we have taken Fig 1 apart and put each figure individually. In this way, the figure of GO Analysis was kept, and the figures of KEGG classification and KOG annotation included supplementary data.
Point 23: (Line 202) Why are the first few words bolded?
Response 23: Thank you for your remind. We have rechecked and revised it.
Point 24: (Table 2) The number of cysteine is less meaningful. The authors may show if their sequences contain the right motif.
Response 24: Thank you for your suggestion. The number of cysteine is one of the basis for judging the type of OBP and CSP, and it is also explained in some articles, (i.e., Li et al. (2021) Comp. Biochem. Physiol. D Genomics Proteomics 40: 100881; Xing et al. (2021) Comp. Biochem. Physiol. D Genomics Proteomics 38: 100814, etc.). Therefore, we kept the description on the number of cysteine. It is true that motif analysis would be better, but the table here is too long and it is too crowded to put in this part. We will definitely conduct motif analysis during the OBP and CSP gene analysis in the subsequent experiments.
Point 25: One species was missed in the figure legend of Fig S4.
Response 25: Thank you for your remind. We have rechecked and revised it.
Point 26: Although both qRT-PCR and RT-qPCR are accepted as abbreviation for quantitative real-time PCR, the authors should choose either one to use throughout the paper.
Response 26: We rechecked the full text and uniformly used qRT-PCR.
Point 27: (Line 110) “…DGEs analysis” should be “…DGE analysis”.
Response 27: Done.
We have provided a point-by-point response to the comments above. Once again, we would like to express our appreciation for your valuable reviews on our manuscript. We hope that the revision can meet the requirements of the journal for publication.
We are looking forward to your reply.
Yours sincerely,
Xiao-Yun Wang
12 June, 2022

Reviewer 2 Report
I have only a limited background in genetics, however, I could see no problems with the scientific aspect of this study, just many editorial comments (listed below) and a couple of concerns with the presentation as my copy of the manuscript shows. The discussion and conclusion sections were strong and overall, well written.
In the study, 76 olfactory-related genes of Glenea cantor Fabricius, a cerambycid beetle, were annotated and identified. This study would lay the groundwork for future research into the mechanism of olfaction of this insect with possible applications into its control, as it is a pest of kapok trees in southern China and southeast Asia. Maybe the common name of this beetle could be given (if the authors know one)?
My editorial comments are as follows:
-line 37, substitute “and” for “which”
-line 47, “pheromones” not “pheromone” (plural vs. singular)
-line 66, “G. cantor is oligophagous and it almost just consume kapok branches.” Is awkwardly worded. I would suggest “G. cantor is oligophagous and consumes almost exclusively kapok branches.”
-lines 69-70, “In Nanning, China, G. cantor is living four generations a year,” I would suggest “In Nanning, China, there are 4 generations of G. cantor a year,”
-line 78, I would suggest inserting “and” before “capture” to make the sentence flow better
-line 87, omit “the”
-line 91, replace “was” with “were”
-lines 145-147, the sentence that begins “To validate the transcriptome,” is incomplete. I can’t suggest an alternate wording as I am not sure what the authors wanted to state.
-line 147, I would suggest “…used as a template…” instead of “used as template”
-line 152, I would add “and” before CSP4.
-lines 152, “…from different age and tissues…” I might suggest “…from differently aged tissues…” but I don’t want to obscure the authors’ meaning.
-line 156, omit “of”
-line 157, “from” instead of “form”
-line 158, “experimental procedure” not “experiment procedure”
-line 159, “tissue’s” instead of “tissues”
-line 165, remove the comma after “analysis”
-line 181, “GO ontologies”; if GO stands for Gene Ontology, do you need to repeat the word ontologies?
-line 187, “most subcategory” do you mean “most common subcategory” or something similar?
-Figure 1; the printing on this figure was too small for me to read in my reviewer’s copy of the manuscript. Maybe enlarge this figure before publishing?
-Section 3.2, lines 201-211, the font is smaller than the rest of the manuscript.
-line 224, substitute “were” for “was”
-line 250, maybe reword this to “Two SNMPs genes were reported and both of them had…”
-line 292, change “provide” to “provides”
-line 293, “lays” instead of “lay”
-line 296, omit “chemical”
-line 311, maybe substitute “so it” for “which” just to make the sentence flow better
-line 327, “detect” instead of “detects”
-line 331, maybe change the wording of this sentence to something like “…and caused the true number of ORs expressed in the antenna in this beetle to be underestimated.” Instead of “…and underestimated the true number of ORs expressed by the antenna in this beetle.”
-line 336, substitute “behaviorally” for “behavior”.
-lines 337-339, this sentence doesn’t read well, in particular “…which indicated that the ORco gene of G. cantor”…indicated what? That it was present? I am reluctant to suggest an alternate wording as I don’t know the authors’ intended meaning.
-line 349, “…proteins, which play…” instead of “…protein, which plays…” (singular / plural issue)
-line 357, omit “While”
-line 362, maybe substitute “interesting” for “interested”
-lines 361-364, this sentence doesn’t read well. As an alternate wording, I would suggest “…while immunohistochemistry is a useful way to locate them in the cells. This could be studied in the future…”
Author Response
Dear Reviewer 2,
We really appreciate your earnest and careful review of our manuscript ID: insects-1763077 entitled as “Antennal transcriptome analysis and identification of olfactory genes in Glenea cantor Fabricius (Cerambycidae: Lamiinae)”. We have revised the manuscript accordingly. Our point-by-point responses are detailed below.
Point 1: line 37, substitute “and” for “which”
Response 1: Done.
Point 2: line 47, “pheromones” not “pheromone” (plural vs. singular)
Response 2: Done.
Point 3: line 66, “G. cantor is oligophagous and it almost just consume kapok branches.” Is awkwardly worded. I would suggest “G. cantor is oligophagous and consumes almost exclusively kapok branches.”
Response 3: Done.
Point 4: lines 69-70, “In Nanning, China, G. cantor is living four generations a year,” I would suggest “In Nanning, China, there are 4 generations of G. cantor a year,”
Response 4: Done.
Point 5: line 78, I would suggest inserting “and” before “capture” to make the sentence flow better
Response 5: Done.
Point 6: line 87, omit “the”
Response 6: Done.
Point 7: line 91, replace “was” with “were”
Response 7: Done.
Point 8: lines 145-147, the sentence that begins “To validate the transcriptome,” is incomplete. I can’t suggest an alternate wording as I am not sure what the authors wanted to state.
Response 8: “To validate the transcriptome, total RNA of mixed age from respectively 10 female antennae and 10 male antennae per replicate (three biological replicates).” has been changed to “To validate the transcriptome, we performed qRT-PCR experiments with the same sequenced RNA samples.”
Point 9: line 147, I would suggest “…used as a template…” instead of “used as template”
Response 9: Done.
Point 10: line 152, I would add “and” before CSP4.
Response 10: Done.
Point 11: lines 152, “…from different age and tissues…” I might suggest “…from differently aged tissues…” but I don’t want to obscure the authors’ meaning.
Response 11: We compared different age and different tissues in the article.
Point 12: line 156, omit “of”
Response 12: Done.
Point 13: line 157, “from” instead of “form”
Response 13: Done.
Point 14: line 158, “experimental procedure” not “experiment procedure”
Response 14: Done.
Point 15: line 159, “tissue’s” instead of “tissues”
Response 15: Done.
Point 16: line 165, remove the comma after “analysis”
Response 16: Done.
Point 17: line 181, “GO ontologies”; if GO stands for Gene Ontology, do you need to repeat the word ontologies?
Response 17: “GO ontologies” has been modified to “GOs”.
Point 18: line 187, “most subcategory” do you mean “most common subcategory” or something similar?
Response 18: “most subcategory” has been changed to “the subcategory annotating the largest number of genes”.
Point 19: Figure 1; the printing on this figure was too small for me to read in my reviewer’s copy of the manuscript. Maybe enlarge this figure before publishing?
Response 19: We have taken Fig 1 apart and put each figure individually in supplementary data.
Point 20: Section 3.2, lines 201-211, the font is smaller than the rest of the manuscript.
Response 20: Thank you for your remind. We have rechecked and revised it.
Point 21: line 224, substitute “were” for “was”
Response 21: Done.
Point 22: line 250, maybe reword this to “Two SNMPs genes were reported and both of them had…”
Response 22: “It was reported that two SNMPs genes were reported and both of them had…” has been rewritten to “Two SNMPs genes were identified and both of them had…”
Point 23: line 292, change “provide” to “provides”
Response 23: Done.
Point 24: line 293, “lays” instead of “lay”
Response 24: Done.
Point 25: line 296, omit “chemical”
Response 25: Done.
Point 26: line 311, maybe substitute “so it” for “which” just to make the sentence flow better
Response 26: Done.
Point 27: line 327, “detect” instead of “detects”.
Response 27: Done.
Point 28: line 331, maybe change the wording of this sentence to something like “…and caused the true number of ORs expressed in the antenna in this beetle to be underestimated.” Instead of “…and underestimated the true number of ORs expressed by the antenna in this beetle.”
Response 28: Done.
Point 29: line 336, substitute “behaviorally” for “behavior”.
Response 29: Done.
Point 30: lines 337-339, this sentence doesn’t read well, in particular “…which indicated that the ORco gene of G. cantor”…indicated what? That it was present? I am reluctant to suggest an alternate wording as I don’t know the authors’ intended meaning.
Response 30: It been rewritten to “GcanOR2 was on the same branch and highly homologous with ORco of other species, so GcanOR2 might be the ORco gene of G. cantor that ORco is a highly conserved receptor found in various insects.”
Point 31: line 349, “…proteins, which play…” instead of “…protein, which plays…” (singular / plural issue)
Response 31: Done.
Point 32: line 357, omit “While”
Response 32: Done.
Point 33: line 362, maybe substitute “interesting” for “interested”
Response 33: Done.
Point 34: lines 361-364, this sentence doesn’t read well. As an alternate wording, I would suggest “…while immunohistochemistry is a useful way to locate them in the cells. This could be studied in the future…”
Response 34: Done.
We have provided a point-by-point response to the comments above. Once again, we would like to express our appreciation for your valuable reviews on our manuscript. We hope that the revision can meet the requirements of the journal for publication.
We are looking forward to your reply.
Yours sincerely,
Xiao-Yun Wang
12 June, 2022

Reviewer 3 Report
This is a very important paper. It is very well written.
This paper represents an incenttive for future projects involving the same subject.
I suggest to include a photo of Glenea cantor to illustrated the species. The illustration represents a vaucher for the science in future.

Author Response
Dear Reviewer 3,
We really appreciate your earnest and careful review of our manuscript ID: insects-1763077 entitled as “Antennal transcriptome analysis and identification of olfactory genes in Glenea cantor Fabricius (Cerambycidae: Lamiinae)”. We have revised the manuscript accordingly. We have compiled your general questions and responded to your suggestions below.
Point 1: I suggest to include a photo of Glenea cantor to illustrated the species. The illustration represents a vaucher for the science in future.
Response 1: Thank you for your suggestion. The photo of Glenea cantor has been added in the supplementary data.
Point 2: Check the genus names of species in the manuscript.
Response 2: Thank you for your suggestion. The species that appear for the first time in this work have written their full names, and we have referred to some articles, such as Li et al. (2021) Comp. Biochem. Physiol. D Genomics Proteomics 40: 100881; Xing et al. (2021) Comp. Biochem. Physiol. D Genomics Proteomics 38: 100814, etc. The genus names of the species after the first appearance are all abbreviated.
Point 3: I suggest to include the author and year of description of each species. The species author must be included only the first time it appears in the text.
Response 3: Thank you for your suggestion. It is definitely better to list the author and year of each species, but in section 2.4, it was only to list the referenced species. If the above information is supplemented, it might appear that this section is not conducive to the reader's reading.
Point 4: “G. cantor is living four generations a year” (line68-69) and “G. cantor has four generations a year” are repeated.
Response 4: Thank you for your remind. “G. cantor has four generations a year” has been removed.
We have provided a point-by-point response to the comments above. Once again, we would like to express our appreciation for your valuable reviews on our manuscript. We hope that the revision can meet the requirements of the journal for publication.
We are looking forward to your reply.
Yours sincerely,
Xiao-Yun Wang
12 June, 2022
